

# HTMC: hierarchical tolerance mask correspondence for human body point cloud registration

Feng Yu, Zhaoxiang Chen, Li Liu, Liyu Ren and Minghua Jiang

School of Computer Science and Artificial Intelligence, Wuhan Textile University, China, Hubei, Wuhan

## ABSTRACT

Point cloud registration can be solved by searching for correspondence pairs. Searching for correspondence pairs in human body point clouds poses some challenges, including: (1) the similar geometrical shapes of the human body are difficult to distinguish. (2) The symmetry of the human body confuses the correspondence pairs searching. To resolve the above issues, this article proposes a Hierarchical Tolerance Mask Correspondence (HTMC) method to achieve better alignment by tolerating obfuscation. First, we define various levels of correspondence pairs and assign different similarity scores for each level. Second, HTMC designs a tolerance loss function to tolerate the obfuscation of correspondence pairs. Third, HTMC uses a differentiable mask to diminish the influence of non-overlapping regions and enhance the influence of overlapping regions. In conclusion, HTMC acknowledges the presence of similar local geometry in human body point clouds. On one hand, it avoids overfitting caused by forcibly distinguishing similar geometries, and on the other hand, it prevents genuine correspondence relationships from being masked by similar geometries. The codes are available at https://github.com/ChenPointCloud/HTMC.

## INTRODUCTION

In recent years, a series of methods (*Mei et al., 2023*; *Salihu & Steinbach, 2023*; *Yang, Shi & Carlone, 2020*) have been proposed to tackle the challenge of point cloud registration task. The human body registration is a special case of point cloud registration. Human body registration offers 3D models for a series of computer vision tasks, such as 3D virtual try-on (*Santesteban et al., 2022*; *Ma et al., 2020*; *Pons-Moll et al., 2017*) and human modeling (*Jiang et al., 2022*; *Li et al., 2022*). In the human body point cloud registration, there are some challenges. In the human body point clouds, there are some similar geometrical shapes in different parts. The similar geometries of human body increase the difficulty to distinguish them. Moreover, the geometrical shape of human body is symmetrical. Some regions naturally have same geometrical shapes. The symmetry of the human body will confuse the feature extractor in the registration pipeline. In addition, the deep learning method relies on the dataset. We produce a human body dataset for point cloud registration.

Corresponding author
Minghua Jiang,
minghuajiang@wtu.edu.cn

There are some remarkable researches in the point cloud registration. Predator (*Huang et al., 2021*) and GeoTansformer (*Qin et al., 2022*) search for correspondence pairs from coarse to fine. RIENet (*Shen et al., 2022*) searches for reliable correspondence pairs. These works handle point cloud registration in various aspects. They explore the correspondence relationship between point pairs using an attention-based approach. However, in human body point clouds, the key concern is how to address the establishment of correspondence relations in the presence of confounding due to similar geometries. To address the issue of confounding correspondence relations due to similar geometry, HTMC acknowledges the presence of them, thereby avoiding genuine correspondence relations from being masked by false ones.

In this article, we propose the Hierarchical Tolerance Mask Correspondence (HTMC) to register the human body point clouds. In point cloud registration, a commonly used pipeline involves searching correspondence pairs, and then using singular value decomposition (SVD) to compute the final rigid transformation. The HTMC builds novel correspondence relations in the human body point clouds. The hierarchical correspondence relations of HTMC allow for the presence of different level correspondence pairs. It assigns different scores to the different level correspondence pairs. The tolerance correspondence relations of the HTMC allow each correspondence pair to have enough similarity rather than precise similarity. The mask of HTMC employs a differentiable mask to make the method focus on the overlapping regions and ignore the non-overlapping region.

In summary, our main contributions can be summarized as follows:

1) We design a hierarchical tolerance correspondence approach tailored to point clouds featuring numerous similar local regions. It allows the presence of similar local regions, preventing genuine correspondence relations from being obscured.
2) We design a differentiable mask for use in correspondence-based registration methods, which intelligently adjusts correspondence relations based on feedback from the results.
3) We produce a human body point cloud dataset specifically designed for training a model for human body point cloud registration.

## BACKGROUND

Point cloud registration aims to find the relative pose relationship between two point clouds, *i.e.*, how one point cloud can be rotated and translated to align with the other point cloud. These two point clouds may be of the same object captured by cameras at different positions, or they may represent an object captured at different moments as it moves. An ancient and classical method (*Besl & McKay, 1992*) involves assuming that the closest points in space correspond to the same point, and then calculating the rotation and translation parameters. Through iterative refinement, it may eventually converge. There are many variations of this method (*Yang, Li & Jia, 2013*; *Segal, Haehnel & Thrun, 2009*; *Bouaziz, Tagliasacchi & Pauly, 2013*) where distance calculation can consider more than just Euclidean distance. Additional information such as normals or other data can be used

to calculate distances. A crucial foundation (*Arun, Huang & Blostein, 1987*) of these methods is that if you know which points in two point clouds correspond to the same point, you can use SVD to calculate the rotation and translation parameters. Two points in different point clouds that correspond to the same point are often referred to as correspondence pair. In reality, points in continuous space are rarely exactly equal, and in this article, correspondence pairs are graded based on the magnitude of their deviations. With the rise of deep learning, an increasing number of methods rely on deep learning techniques to find correspondence pairs. These methods focus on mitigating the interference of rotation and translation on point cloud features. The goal is to ensure that rotation and translation do not change the point features, allowing the identification of correspondence pairs after rotation. However, this approach may cause the network to overlook symmetric differences. In human body point clouds, there are many symmetric regions, and it's essential to consider the presence of these symmetric areas when registering human body point clouds.

## RELATED WORK

Our work focuses on registering partially overlapping human body point clouds through the method of finding correspondence pairs. Therefore, we mainly introduce partial overlap registration and correspondence-based registration methods.

### Partial overlapping registration

The partial overlapping registration aligns two point clouds that have some points without corresponding points in another point cloud. The OMNet (*Xu et al., 2021*) uses the mask to reduce the influence of the outliers in feature embedding for partial overlapping registration. RPMnet (*Yew & Lee, 2020*) introduces the slack variables to allow for points to have no correspondence point in another point cloud. Structure-based Overlap Matching (STORM) (*Wang et al., 2022*) uses an overlap prediction module to register partial overlapping point clouds. Those methods make a remarkable contribution to partial overlapping registration. Due to the symmetry of human body point clouds, the masking mechanism may mistakenly remove overlapping regions, resulting in missing inliers during registration. On the other hand, the slack variable requires the learned features to be sufficiently discriminative, but the presence of numerous similar geometries in human body point clouds can weaken the distinctiveness between features. In our work, we use a mask mechanism adjusting the similarity scores to concentrate on overlapping regions and use an evaluator to limit the negative influence of the errors in the mask.

### Correspondence-based registration

In point cloud registration, there is a common method where correspondence pairs are first identified, and then a rigid transformation is computed using the mathematical method SVD. Similar approaches can also be referred to as correspondence-based registration methods. The Iterative Closest Point (ICP) (*Besl & McKay, 1992*) builds correspondence pairs by spatial distance and iteratively updates them. Iterative distance-aware similarity matrix convolution (IDAM) (*Li et al., 2020*) also uses spatial distance

information, which is different from ICP in that it uses it to compute features through the network and then find correspondence pairs through the features. RIEnet (*Shen et al., 2022*) refines the correspondence relations and evaluates the inliers to obtain the robust correspondence pairs. *Huang et al. (2022)* uses coarse-to-fine strategy to establish the correspondence relations. The deep closest point (DCP) (*Wang & Solomon, 2019*) is a special correspondence-based method. It sends the deep closest point rather than real point to SVD to register point cloud. The correspondence-based methods register point clouds by building reliable correspondence relations or generating deep closest points. In searching for correspondence pairs in human body point clouds, it is essential to consider the presence of similar geometries. Our method establishes a novel correspondence relation to address the existence of similar geometries, aiming to achieve higher registration accuracy.

## METHODOLOGY

In this section, we will briefly introduce the motivation behind the design of HTMC and its advantages in human body registration. Moreover, we will describe the details of our whole pipeline.

### Problem description

Point cloud registration aims to find a rigid transformation that aligns two point clouds. Specifically, it involves searching for a rotation matrix and a translation vector that minimizes the Euclidean distance between correspondence pairs. It can be described as follows:

$$R, t = \arg\min_{R,t} \sum_{x_i, y_j} ||Rx_i + t - y_i|| \tag{1}$$

where $(x_i, y_i)$ denotes one of searched correspondence pairs. $R$ denotes rotation matrix. $t$ denotes translation vector. Equation (1) can be solved using the mathematical method SVD. The objective of the correspondence-based method is to find some correspondence pairs $(x_i, y_j)$.

### HTMC overall architecture

The HTMC is designed to register human body point clouds. The HTMC defines different level correspondence pairs to acknowledge the presence of similar geometries in human body point clouds, and uses tolerance methods to set the optimization goals for strict corresponding, approximate corresponding and non-corresponding pairs during training. In addition, the human body is partially overlapping. HTMC uses a differentiable mask to focus on the overlapping region.

Figure 1 depicts the details of correspondence relation calculation in HTMC. To utilize both the global and local message, we follow the dynamic graph CNN (DGCNN) (*Wang et al., 2019*) to construct edges as the initial input of the network. This operation concatenates the k nearest neighbors for each point. The HTMC uses shared weight multilayer perceptron (MLP) (*Qi et al., 2017*) for feature embedding. We first extract point-wise features (*fx* and *fy*) for two point clouds, respectively. Then we fuse two point
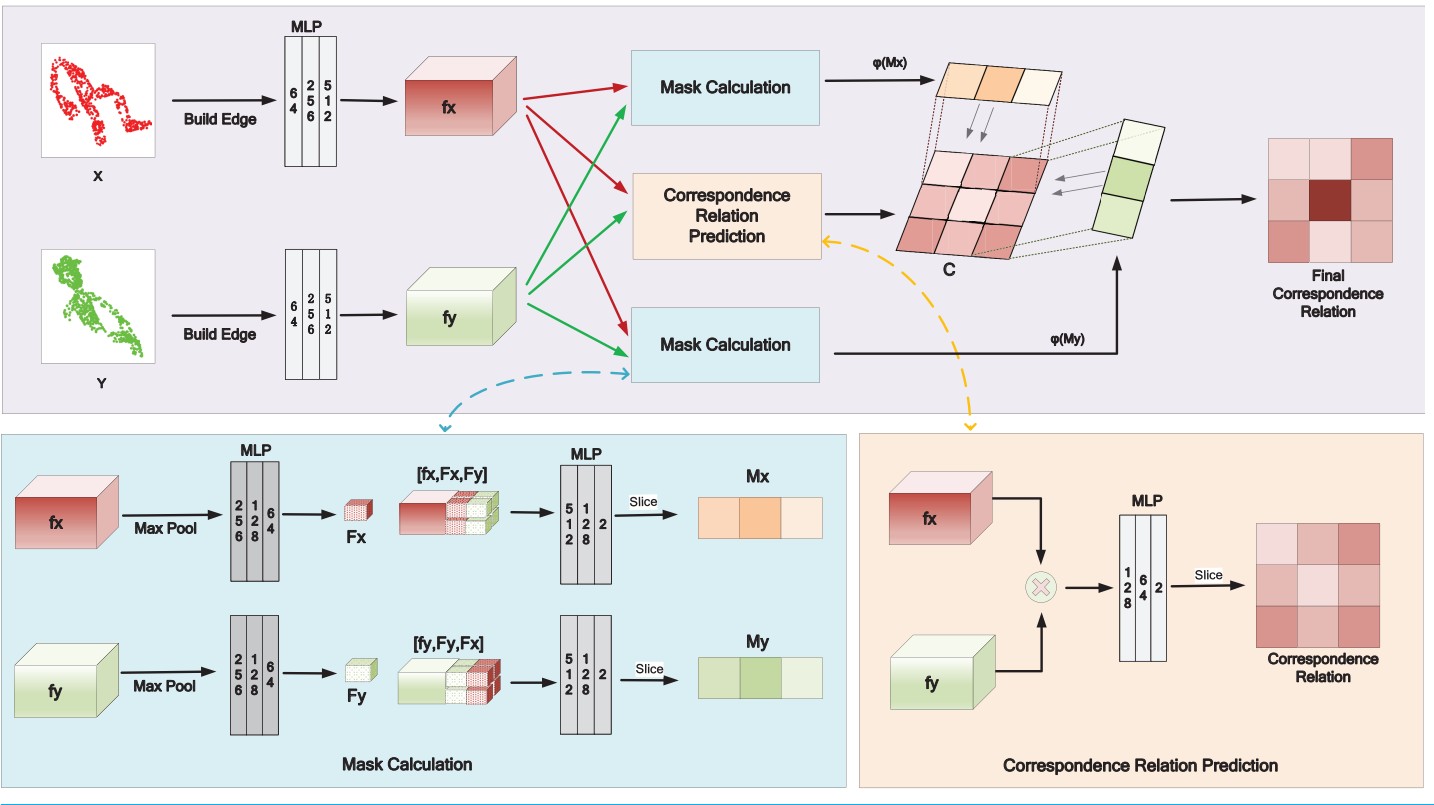

**Figure 1  Calculation of correspondence relation in our pipeline.**     

cloud features and then use an MLP to predict initial correspondence relations *C*. We use max pooling and MLP to aggregate point-wise features into global features (*Fx* and *Fy*). Then, the point-wise features are concatenated with the global feature of two point cloud to potentially describe its position in the two point clouds. Then, HTMC uses MLP to learn the inlier masks (*Mx* and *My*) and applies a mapping function $\phi()$ to map the masks. Finally, the initial correspondence relation *C* multiplies with two mapped masks to obtain the final correspondence relation.

After obtaining the correspondence relation, HTMC constructs the correspondence pairs based on the relation. Then, HTCM filters out wrong correspondence pairs based on spatial information. Finally, SVD is used to compute the final rigid transformation.

## Hierarchical tolerance correspondences

In traditional deep learning methods (*Pais et al., 2020*; *Bai et al., 2021*; *Fu et al., 2021*), the point pairs are classified into two categories: corresponding or not corresponding. However, there are many similar geometries in the human body point cloud. Treating the human body correspondence pairs searching as a binary classification problem is not enough. In this work, HTMC converts the binary classification problem into a hierarchical problem.

We first introduce how to define different level correspondences. In the point cloud registration problem, the points in the same position after two point cloud aligned are the same points. We call the point pair that consists of same points as a correspondence pair. However, the same position is abstract for computers, and it needs a more concrete definition. In general, two points whose distance is less than a certain value can be considered as being in the same position. In other works, they choose one value as threshold to define whether two points are in the same position. In our work, we choose three variable values to define three levels of correspondence. The first level correspondence, which is strict corresponding point pairs, is defined by the distance being less than half of the average nearest neighbor distance (ANND). The other two levels correspondences are less than ANND and 1.5 times ANND, respectively. They are approximately corresponding.

Then, we introduce why they are tolerant. In traditional methods, the cross-entropy function is used for binary classification problems. The cross-entropy function hopes that the predicted scores are closer to zero or one. However, due to the presence of similar geometries in human body, the approximately corresponding pairs also exhibit similarities. Forcing them to be close to zero or one is not reasonable. Therefore, we assign varying default similarity scores for different levels of correspondences. And we only require that the predicted scores meet their respective level rather than being close to the default similarity scores.

## Differentiable mask

The human body registration task is a partial overlapping point cloud registration task. HTMC uses differentiable mask module to concentrate on the overlapping regions and reduce the influence of the non-overlapping regions. Generally, the mask should consist of zeros and ones that indicate whether a point is located in the overlapping region. However, discrete values are not friendly for gradient backpropagation. Moreover, different points may hold varying degrees of importance for building correspondence relations. Merely assigning them as two zeros or ones will lose some information.

To facilitate gradient backpropagation and retain more information, HTMC uses a function $\phi()$ to map the mask to adjust the correspondence relations. Specifically, each point pair consists of two points, its similarity scores multiply with two mapped masks of two points to obtain a new similarity score of this point pair. The similarity scores describe the correspondence relation between two points. HTMC learns masks from point features and two point cloud global features. The mask consists of continuous values between zero and one. Those values are less than one, similarity scores directly multiplying those values will always reduce the similarity scores. The mapping function $\phi()$ is used to map the mask values. The mapped mask will increase the similarity scores of point pairs if their points are located in overlapping region, and reduce the similarity scores of non-overlapping region point pairs.

## Registration evaluator

Due to the symmetry and the presence of similar geometries of human body, the mask predicted by the model may contain some errors. If it treats the non-overlapping region as an overlapping region, it will introduce more wrong correspondences. And if it treats the overlapping region as a non-overlapping region, it will eliminate the correct correspondences. To increase the robustness of the HTMC, a registration evaluator is used to choose the best registration scheme. The registration evaluator has two functions, it uses different schemes to register human body point cloud and selects the best one. The mask has two usages, it can increase the similarity scores in the overlapping region, and reduce the similarity scores in the non-overlapping region. Therefore, we can have four schemes to utilize the mask. A balance scheme both increases the similarity scores in overlapping regions and reduces similarity scores in non-overlapping regions. If we want to explore more potential correspondences, we can only increase the similarity scores in overlapping region. If we want to obtain more precise correspondences, we can only reduce the similarity scores in non-overlapping regions. To avoid the wrong mask causing registration failure, we can keep the origin scores to avoid the influence of the wrong mask.

The above four schemes will result in four rigid transformations. HTMC transforms the target point cloud to align with the source point cloud based on four transformations and calculates the number of corresponding points after alignment. The result that has the most corresponding points can be considered as the final result.

## Loss function

In the human body point cloud, there are some similar geometries. The HTMC uses hierarchical tolerance correspondences to address this issue. The loss functions of HTMC are designed to learn those correspondences and inlier masks. The loss of strict correspondences can be described as follows:

$$L_{st} = \sum \left( ReLu(C_i - l_1) \right)^2 \tag{2}$$

where $C_i$ denotes the predicted similarity score of each strict corresponding pair, $l_1$ denotes the default score of first level correspondences. The first level correspondences are strictly corresponding, and the others are approximately corresponding.

Unlike the classical cross-entropy function to force the predicted results closer to the ground truth, we use $ReLu()$ to allow the predicted scores to meet their levels.

The loss of approximate correspondences can be described as follows:

$$L_{ap} = k \sum \left( (ReLu(l_2 - C_i))^2 + (ReLu(l_3 - C_j))^2 \right) \tag{3}$$

where $C_i$ denotes the predicted similarity score of each second level corresponding pair, $C_j$ denotes the predicted similarity score of each third level correspondence pairs, $l_2$ and $l_3$ denote the default scores of second and third level correspondences, respectively. Since the point pairs are more than strict corresponding pairs, $k$ is used to balance their influences.

The loss of non-corresponding pairs can be described as follows:

$$L_{no} = \mathrm{k} \sum \left( ReLu(l_{no} - C_i) \right)^2 \tag{4}$$

where $C_i$ denotes the predicted similarity scores of non-corresponding pairs, $l_{no}$ denotes the default scores of non-corresponding pairs.

The mask loss can be described as follows:

$$L_{mask} = H(M_{pre}, M_{gt}) \tag{5}$$

where $H()$ denotes the cross-entropy function, $M_{pre}$ denotes the predicted mask, and the $M_g t$ denotes the ground truth mask.

The final loss is the sum of the above losses, it can be described as follows:

$$L = L_{st} + L_{ap} + L_{no} + L_{mask} \tag{6}$$

# EXPERIMENT

## Experiment environment

We implement HTMC on Windows 10 using the NVIDIA GeForce RTX 3090Ti and Intel (R) Core(TM) i9-12900KF CPU. Our experiments use PyTorch for implementation. We use ADAM (*Kingma & Ba, 2014*) as an optimizer. The total number of epochs is set to 35. The initial learning rate is set to 0.001, and it multiplies 0.1 at epoch 15, 30.

## Dataset

We produce a human body dataset for training a human body registration network. This dataset contains 5,792 point clouds, and we use 4,500 point clouds for training and 1,292 point clouds for testing. It comprises three processing steps. The first step involves downsampling to obtain point clouds with varying densities, which results in different point cloud densities within the entire dataset. The second step involves clipping the point clouds to obtain partially overlapping regions. This creates non-overlapping areas between the two point clouds, which can be important for the network to learn to handle symmetric differences. For instance, one point cloud might lack a right hand, while the other might lack a left hand. This step is crucial for the network to handle such situations. The third step involves applying rotation and translation to the point clouds, serving as a data augmentation technique that provides the network with theoretically infinite variations of the same samples. For specific processing details, you can refer to the provided source code. After the processing, each point cloud has 768 points inputted into the network. Each point cloud has 30–80% inlier points.

## Evaluate metric

In this experiment, we use root mean square error (RMSE) and mean absolute error (MAE) (*Yew & Lee, 2020*) to evaluate the rotation and translation errors. The RMSE and MAE can be described as follows:

$$RMSE(R) = \sqrt{\frac{1}{n}\sum\frac{1}{3}\sum(R_{gt} - R_{pre})^2} \tag{7}$$

$$RMSE(T) = \sqrt{\frac{1}{n}\sum\frac{1}{3}\sum(T_{gt} - T_{pre})^2} \tag{8}$$

$$MAE(R) = \sqrt{\frac{1}{n}\sum\frac{1}{3}\sum|R_{gt} - R_{pre}|} \tag{9}$$

$$MAE(T) = \sqrt{\frac{1}{n}\sum\frac{1}{3}\sum|(T_{gt} - T_{pre})|}. \tag{10}$$

We also define the bias of rotation less than 1 and the bias of translation less than 0.1 as successful registration to calculate the registration recall (RR). The bias is calculated as follows:

$$bias(R) = \sqrt{\sum(R_{gt} - R_{pre})^2} \tag{11}$$

$$bias(T) = \sqrt{\sum(T_{gt} - T_{pre})^2}. \tag{12}$$

In addition, to evaluate the quality of the found correspondence pairs, we also calculate the accuracy, recall, precision, and F1-score of strictly corresponding pairs. Table 1 shows the metrics employed by our method and other deep learning methods. While other methods primarily assess registration quality, we aim to counteract the interference in correspondence pairs search due to the symmetry and similar local structures in human body point clouds. Therefore, we have included an additional evaluation of the quality of correspondence pairs.

## Comparison with other methods

In this experiment, we compare the HTMC with other methods to evaluate its performance. HTMC is compared with the classical traditional method ICP (*Besl & McKay, 1992*) and correspondence-based deep learning methods DCP (*Wang & Solomon, 2019*), IDAM (*Li et al., 2020*) and RIENet (*Shen et al., 2022*) to evaluate its performance. The visualization of the results of HTMC and other methods is shown in Fig. 2. The more detailed performance metrics are shown in Table 2. As shown in Table 2, the other deep learning methods lack the ability to identify the correspondence pairs in the human body dataset. Their registration results are essentially random. The ICP and IDAM can successfully register some point clouds, but their performance is not stable. The HTMC can successfully establish the correspondence relations to register most point clouds.

Due to the HTMC acknowledging the presence of similar geometries, it may introduce some wrong correspondence pairs. Furthermore, the mask mechanism will increase the similarity scores of overlapping regions, it also brings some wrong correspondence pairs. Therefore, HTMC has a low precision rate for strict corresponding pairs, but it has a high recall rate for strict corresponding pairs. In theory, only three correspondence pairs are needed for successful registration in our registration pipeline. However, too many correspondence pairs will confuse the filter, and insufficient correspondence pairs will

**Table 1 The used metrics of HTMC and other methods.**

|  | RMSE (R) | MAE (R) | RMSE (t) | MAE (t) | Registration recall | Accuracy | Recall | Precision | F1-score |
|---|---|---|---|---|---|---|---|---|---|
| DCP | ✓ | ✓ | ✓ | ✓ |  |  |  |  |  |
| IDAM | ✓ | ✓ | ✓ | ✓ |  |  |  |  |  |
| RIE | ✓ | ✓ | ✓ | ✓ |  |  |  |  |  |
| Ours | ✓ | ✓ | ✓ | ✓ | ✓ |  | ✓ | ✓ | ✓ | ✓ |

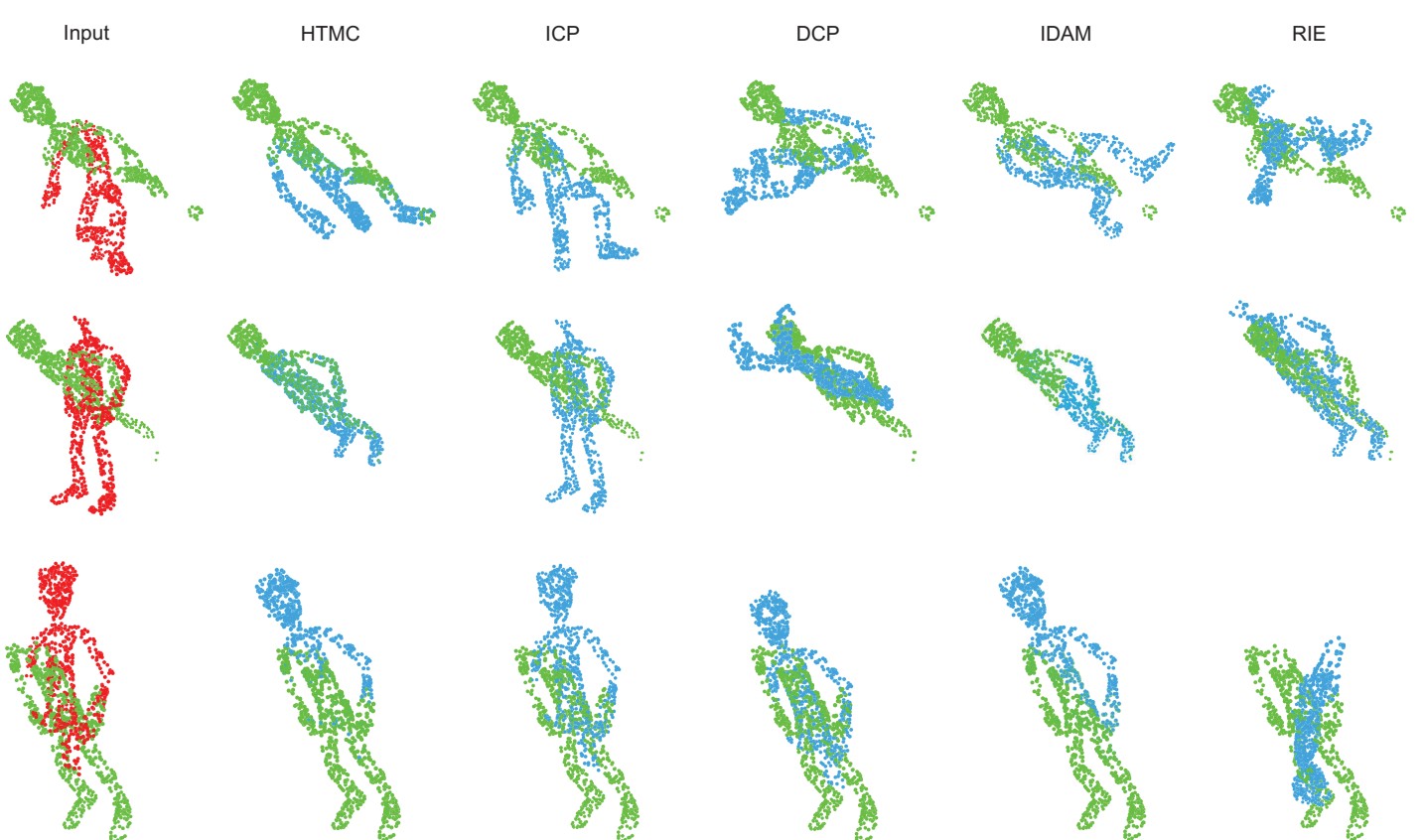

**Figure 2 Visualization of the registration results of HTMC and other methods.** The red point clouds denote source point clouds. The green point clouds denote the target point cloud. The blue point clouds denote the transformed source point cloud.

**Table 2 The performance of HTMC and other methods.** Bold indicates the best performance in the current metric.

|  | RMSE (R) | MAE (R) | RMSE (T) | MAE (t) | RR | Accuracy | Recall | Precision | F1-score |
|---|---|---|---|---|---|---|---|---|---|
| ICP (*Besl & McKay, 1992*) | 22.9503 | 18.3183 | 1.8304 | 1.2506 | 0.0604 | – | – | – | – |
| DCP (*Wang & Solomon, 2019*) | 82.8013 | 61.3060 | 7.0631 | 5.4069 | 0.0000 | 0.9963 | 0.0001 | 0.0088 | 0.0002 |
| IDAM (*Li et al., 2020*) | 17.9036 | 8.9468 | 3.6148 | 1.8054 | 0.0650 | 0.9841 | 0.1895 | **0.6863** | 0.2970 |
| RIENet (*Shen et al., 2022*) | 58.0840 | 38.2578 | 19.0655 | 15.3257 | 0.0000 | 0.9961 | 0.0115 | 0.3014 | 0.0221 |
| Ours | **0.4236** | **0.0183** | **0.0941** | **0.0037** | **0.9938** | **0.9988** | **0.8376** | 0.4183 | **0.5580** |

**Table 3 The performance of HTMC on ModelNet40.** Bold indicates the best performance in the current metric.

| Model | ModelNet40 | | | | | | | | |
|---|---|---|---|---|---|---|---|---|---|
| | RMSE (R) | MAE (R) | RMSE (t) | MAE (t) | Registration recall | Accuracy | Recall | Precision | F1-score |
| ICP | 24.022577 | 16.369591 | 0.193026 | 0.145436 | 0.0372 | – | – | – | – |
| DCP | 6.409776 | 4.555631 | 0.033049 | 0.025002 | 0.0028 | 0.9911 | <0.0001 | nan | nan |
| IDAM | 3.672205 | 0.908155 | 0.021015 | 0.005480 | 0.6863 | 0.9691 | 0.1675 | 0.9155 | 0.2832 |
| RIE | 0.009396 | 0.001947 | 0.000051 | 0.000016 | **1.0000** | 0.9886 | 0.0998 | 0.9616 | 0.1808 |
| HTMC | **0.001430** | **0.000033** | **0.000005** | **<0.000001** | **1.0000** | **0.9998** | **0.9158** | **0.9621** | **0.9384** |

**Table 4 The performance of HTMC on KITTI.** Bold indicates the best performance in the current metric.

| Model | KITTI | | | | | | | | |
|---|---|---|---|---|---|---|---|---|---|
| | RMSE (R) | MAE (R) | RMSE (t) | MAE (t) | Registration recall | Accuracy | Recall | Precision | F1-score |
| ICP | 8.504693 | 4.255252 | 0.409459 | 0.210804 | 0.480968 | – | – | – | – |
| DCP | 106.295723 | 85.591461 | 3.266513 | 2.507504 | 0.000000 | 0.9965 | 0.0000 | nan | nan |
| IDAM | 25.709606 | 8.951884 | 1.446016 | 0.660434 | 0.235672 | 0.9808 | 0.1575 | **0.6267** | 0.2517 |
| RIE | 102.234161 | 80.942940 | 4.920786 | 3.820946 | 0.000000 | 0.9966 | 0.0001 | 0.0054 | 0.0002 |
| HTMC | **2.296907** | **1.243306** | **0.080559** | **0.057598** | **0.762578** | **0.9995** | **0.5996** | 0.4574 | **0.5189** |

increase the risk of registration failure. In this experiment, the recall and precision remain within a safe range, therefore HTMC will have high registration recall and low error of rotations and translations.

## Comparison evaluation on ModelNet40

To further demonstrate the performance of our method, we compared it with other methods on public datasets. Following RIENet (*Shen et al., 2022*), we conducted experiments on the ModelNet40 (*Wu et al., 2015*) dataset.

ModelNet40 is a relatively simple dataset. As shown in Table 3, both our method and other approaches have achieved good performance in point cloud registration. However, the difference is that our method is able to find more correspondence pairs. Traditional methods assume that the correspondence relations between points are one-to-one, making them susceptible to being deceived by spurious correspondence pairs. In contrast, our method utilizes a hierarchical tolerance approach to explore more potential correspondence relations.

## Comparison evaluation on KITTI

The KITTI dataset (*Geiger, Lenz & Urtasun, 2012*) is an outdoor dataset used in autonomous driving. We utilized the preprocessed KITTI dataset provided by DeTarNet (*Chen, Yang & Tao, 2022*) and performed additional processing on it. In this work, to demonstrate our superiority in exploring correspondence pairs, we performed random downsampling to reduce the resolution and removed contiguous overlapping regions to

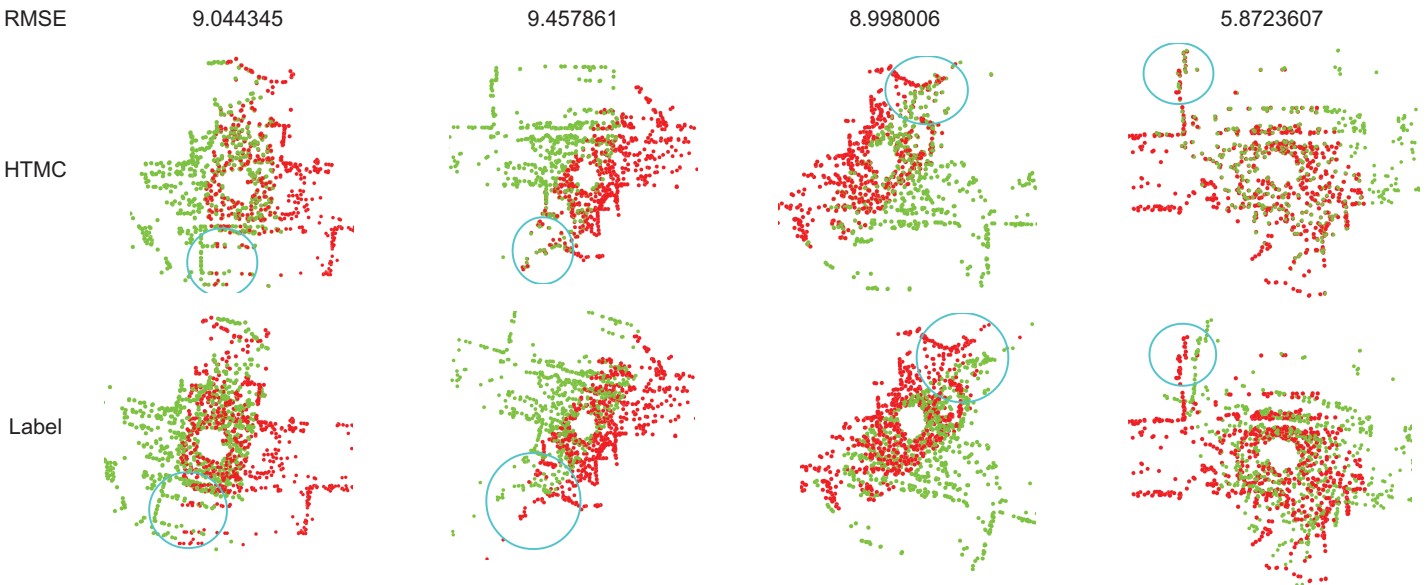

**Figure 3** The recall of correspondence pairs of detaching and attaching the mask to learn correspondence relationships during training.

lower the overlap rate of the point cloud. After processing, each point cloud contains 768 points, with only 30–80% of the points being inliers. The performance of HTMC on KITTI is shown in Table 4.

The processed KITTI dataset contains only a small number of inliers, and other algorithms suffer from performance degradation in this scenario. In contrast, our method maintains stability. The original KITTI dataset is a large-scale point cloud, and its inliers are not perfectly corresponded. At low resolution, the bias between its inliers becomes larger. This deviation limits the upper bound of accuracy. Additionally, the original KITTI dataset contains inaccurate labels. This can result in correctly registered samples having relatively large error values when calculating the rotation error. Figure 3 illustrates some samples with errors caused by incorrect labels. From the circular parts in the graph, it can be observed that HTMC's registration performance is better than the labels provided by the dataset. However, due to the inaccuracy of the dataset labels, correctly registered results may report large rotation errors.

## Ablation study

In this section, we conduct some ablation studies to verify the effectiveness of the hierarchical correspondence and the mask mechanism.

### Study on hierarchical correspondences

We remove the mask mechanism and replace the hierarchical correspondences with the traditional zero-one correspondences as the base model. To analyze the impact of different default similarity scores on registration results, we design three types of default similarity scores. The scores of non-corresponding and strictly corresponding pairs are set to 0.1 and

**Table 5 The performance of three types of hierarchical correspondences and base model.** Bold indicates the best performance in the current metric.

| | RMSE (R) | MAE (R) | RMSE (T) | MAE (t) | RR | Accuracy | Recall | Precision | F1-score |
|---|---|---|---|---|---|---|---|---|---|
| Base | 12.8703 | 4.5939 | 4.5718 | 1.8182 | 0.4389 | 0.9990 | 0.1030 | 0.2976 | 0.1530 |
| Strict | 5.6072 | 0.9398 | 1.5924 | 0.2490 | 0.9187 | 0.9993 | 0.4592 | 0.6703 | 0.5450 |
| Balance | 3.7358 | 0.4392 | 1.3234 | 0.1306 | **0.9598** | **0.9995** | 0.5947 | **0.7304** | **0.6556** |
| Lenient | **3.5528** | **0.4298** | **1.2322** | **0.1136** | 0.9582 | 0.9994 | **0.6017** | 0.7044 | 0.6490 |

0.9, respectively. The first type is a strict scheme that focuses on identifying approximate corresponding pairs, its $l2$ and $l3$ are set to 0.5 and 0.2. The second type is a balance scheme, its $l2$ and $l3$ are set to 0.6 and 0.3. The third type is a lenient scheme that allows approximate corresponding pairs similar to strict corresponding pairs, its $l2$ and $l3$ are set to 0.8 and 0.5.

As shown in Table 5, the base model fails to build robust correspondence relations. It only has 43.8854% registration recall. After applying the hierarchical correspondences, the recall and precision of correspondence pairs have significant improvement. Due to the similar geometries in human body, the strict scheme will confuse the model. Both the recall and precision of correspondence pairs are inferior to others. The balanced scheme has the best precision of correspondence pairs and registration recall. However, it has a higher transformation error, which indicates that it fails to register some samples. Note that no successful registration is different from registration failure. No success samples may have more bias than the requirement, but the failure samples indicate that their results are random results. The lenient scheme will find more correspondence pairs but it also introduces more fake correspondence pairs.

*Study on mask mechanism*

In this experiment, we use the lenient hierarchical correspondences model as the base model. As described in "Differentiable Mask", we need a mapping function $\phi()$ to map the mask. To analyze the impact of different mapping functions $\phi()$ on registration results, we design three types of mapping function $\phi()$. The first type of $\phi()$ is the root function, it can be described as follows:

$$\phi(x) = \frac{\sqrt{|x - 0.5|}}{\sqrt{0.5 * 2}} * \frac{|x - 0.5|}{x - 0.5} + 1 (x \in [0, 0.5) \cup (0.5, 1]). \tag{13}$$

In fact, the main part of this function is $\sqrt{|x - 0.5|}$. The $\frac{|x-0.5|}{x-0.5} * \sqrt{|x - 0.5|}$ guarantees the function having definition in interval $[0,0.5)$. Note that it is no definition in 0.5. However, it rarely strictly equals to 0.5 in computers, and losing a few point pairs scores is acceptable. The $\frac{\sqrt{|x-0.5|}}{\sqrt{0.5*2}}$ guarantees the difference between maximum and minimum values of function equaling to one. Due to the difference between maximum and minimum values of the original mask being equal to one, this component ensures that the difference between maximum and minimum values of three map functions is also one, thereby

**Table 6 The performance of three types of mapping functions and base model.** Bold indicates the best performance in the current metric.

| | RMSE (R) | MAE (R) | RMSE (T) | MAE (t) | RR | Accuracy | Recall | Precision | F1-score |
|---|---|---|---|---|---|---|---|---|---|
| Base | 3.5528 | 0.4298 | 1.2323 | 0.1136 | 0.9582 | 0.9994 | 0.6017 | 0.7044 | 0.6490 |
| None | 2.2536 | 0.2567 | 0.7863 | 0.0735 | 0.9721 | **0.9995** | 0.6726 | **0.7575** | **0.7125** |
| Root | **0.2215** | **0.0102** | **0.0611** | **0.0025** | 0.9946 | 0.9988 | 0.8323 | 0.4057 | 0.5455 |
| Linear | 0.4236 | 0.0183 | 0.0941 | 0.0037 | 0.9938 | 0.9988 | **0.8376** | 0.4183 | 0.5580 |
| Square | 0.5889 | 0.0313 | 0.1445 | 0.0059 | 0.9907 | 0.9988 | 0.8375 | 0.4346 | 0.5722 |

facilitating a better comparison among the three functions. The $+1$ guarantees the function greater than one when $x$ is greater than 0.5, and less than one when $x$ is less than 0.5. It can make the point pairs have higher similarity scores when it contains overlapping region points, and have lower similarity scores when it contains non-overlapping region points. The second type of $\phi()$ is a linear function, it can be described as follows:

$$\phi(x) = \frac{|x - 0.5|}{0.5 * 2} * \frac{|x - 0.5|}{x - 0.5} + 1 = x + 0.5(x \in [0, 0.5) \cup (0.5, 1]) \tag{14}$$

The main part of this function is $x - 0.5$, the other parts are explained above. Note that this equation is not hold strictly, we ignore that it has no definition in 0.5. The third type of $\phi()$ is a square function, which can be described as follows:

$$\phi(x) = \frac{|x - 0.5|^2}{0.5^2 * 2} * \frac{|x - 0.5|}{x - 0.5} + 1(x \in [0, 0.5) \cup (0.5, 1]) \tag{15}$$

The main part of this function is $(x - 0.5)^2$.

As shown in Table 6, loading the mask mechanism can significantly improve the performance of the model. The 'None' model denotes $\phi(x) = 1$. It trains the mask but does not use it. As shown in Fig. 1, the mask module and correspondence relation module share the feature extractor to extract point-wise features. Therefore, training the mask may improve the performance of this extractor. The experiment shows that the model has higher accuracy, recall, and precision rate of correspondence pairs compared to the base model. It indicates that a training mask is beneficial for building correspondence relations even if the trained mask is not used. After applying the mask, the model has higher recall and lower registration errors. The more correspondence pairs can reduce the risk of registration failure, the lower RMSE of rotation and translation indicates that it has few registration failure samples. In general, introducing more fake correspondence pairs can confuse the network. However, our mask mechanism does not have this defect. As described in "Registration Evaluator", we use registration evaluator to choose the best scheme. If the fake correspondence pairs worsen the registration performance, the evaluator will choose other schemes. Therefore, it can achieve better registration performance even if it has lower precision of correspondence pairs after applying the mask mechanism. As shown in Table 6, the root map function exhibits the best registration performance, and the square map function exhibits the best correspondence relations. This

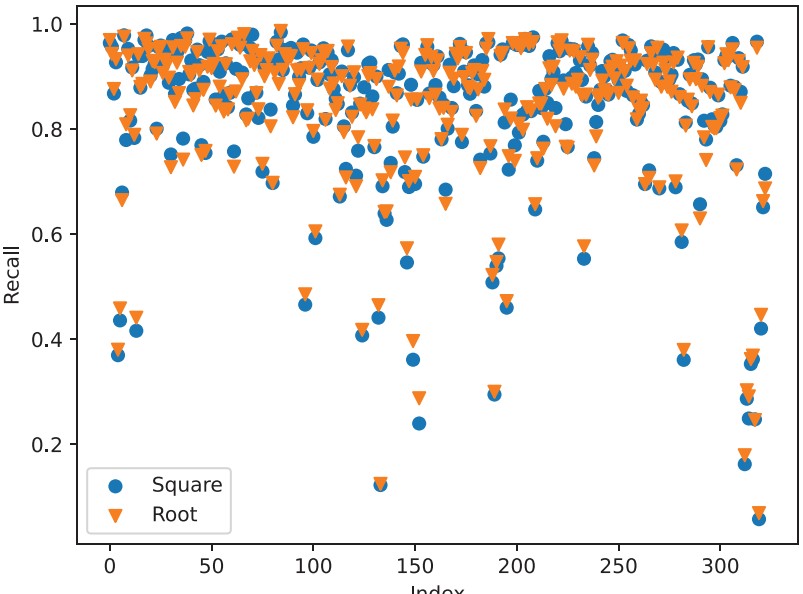

**Figure 4** The recall of correspondence pairs in each batch samples of the square map function and the root map function.

result is abnormal, better correspondence relations should result in better registration performance. To better understand its reason, we check the recall of correspondence pairs in each batch of samples. From Fig. 4, we can find out why the square map function has the best registration performance. Despite the total recall of the root map function is lower than the square map function, the root map function has higher recall when the recall is low. When the network finds enough correspondence pairs, additional correspondence pairs may not have a significant effect. However, if the network only finds out a few correspondence pairs, more correspondence pairs are more likely to result in a significant improvement in registration performance.

In summary, the root map function is the best map function. However, those functions have no significant difference. For the simple consideration, we choose the linear function as the map function of HTMC.

### The influence of mask mechanism

The study on mask mechanism discovers that training maps can help obtain a robust feature extractor. However, it requires the correct training method. Training the mask does not always bring a positive impact. As described in Fig. 1, the mask is used to adjust the correspondence relation matrix. Therefore, there are two correspondence relations. During the training step, we should pick one to calculate the correspondence loss. If we attach the mask to learn correspondence relations, it will use the adjusted correspondence relation matrix to calculate correspondence loss, otherwise, it uses the original correspondence relation matrix. Figure 5 shows the difference between attaching and detaching the mask to learn correspondence relations. As shown in Fig. 5, the network is underfitting when attaching a mask to learn correspondence relations. As described in "Study on mask

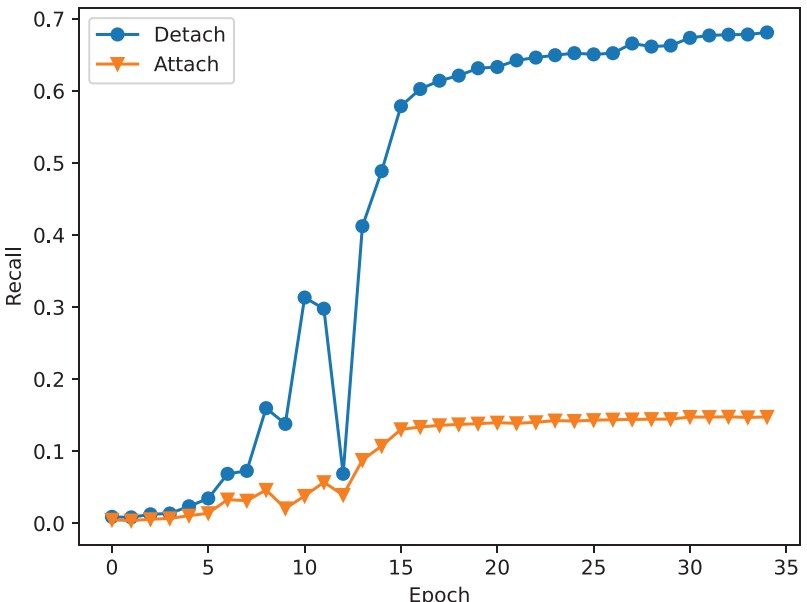

**Figure 5 The recall of correspondence pairs of detaching and attaching the mask to learn correspondence relation during training.**

mechanism", applying the mask will introduce a lot of fake correspondence pairs. It will confuse the network during the training stage. In summary, the mask mechanism is useful. It can help to train a robust feature extractor because it shares the feature extractor with the correspondence relation prediction module. However, it can also confuse the network, so it should be detached from learning correspondence relations during the training stage.

## CONCLUSION

In this work, we proposed HTMC to register human body point clouds and produce a human body dataset. HTMC uses hierarchical correspondences to acknowledge the presence of similar geometries in human body point clouds. It increased the registration success rate of HTMC in human body point clouds from 43.89% to 95.82%. We also designed a differentiable mask mechanism to reduce the influence of symmetrical parts of human body in non-overlapping regions. It further elevated the registration success rate to 99.88%. The experiment results indicate that HTMC establishes robust correspondence relations to successfully register more than 99% human body point clouds.

### Funding

This work was supported by the National Natural Science Foundation of China (No. 62202346), the Hubei Key Research and Development Program (No. 2021BAA042), the Open Project of Engineering Research Center of Hubei Province for Clothing Information (No. 2022HBCI01), the Wuhan Applied Basic Frontier Research project (No. 2022013988065212), MIIT's AI Industry Innovation Task unveils flagship projects (Key

technologies, equipment, and systems for flexible customized and intelligent manufacturing in the clothing industry), and the Hubei Science and Technology Project of Safe Production Special Fund (Scene control platform based on proprioception information computing of artificial intelligence). The funders had no role in study design, data collection and analysis, decision to publish, or preparation of the manuscript.

## Grant Disclosures

The following grant information was disclosed by the authors:

National Natural Science Foundation of China: 62202346.

Hubei Key Research and Development Program: 2021BAA042.

Open Project of Engineering Research Center of Hubei Province for Clothing Information: 2022HBCI01.

Wuhan Applied Basic Frontier Research Project: 2022013988065212.

MIIT's AI Industry Innovation Task Unveils Flagship Projects.

Hubei Science and Technology Project of Safe Production Special Fund.

## Competing Interests

The authors declare that they have no competing interests.

## Author Contributions

- Feng Yu conceived and designed the experiments, authored or reviewed drafts of the article, and approved the final draft.
- Zhaoxiang Chen conceived and designed the experiments, performed the experiments, analyzed the data, performed the computation work, prepared figures and/or tables, and approved the final draft.
- Li Liu analyzed the data, prepared figures and/or tables, authored or reviewed drafts of the article, and approved the final draft.
- Liyu Ren performed the experiments, analyzed the data, performed the computation work, prepared figures and/or tables, and approved the final draft.
- Minghua Jiang conceived and designed the experiments, prepared figures and/or tables, authored or reviewed drafts of the article, and approved the final draft.

## Data Availability

The experimental results can be obtained using the source code in the Supplemental File.

The ICP codes are available at open3D Library: http://www.open3d.org/. The ModelNet40 dataset are available at: https://3dshapenets.cs.princeton.edu/. The KITTI dataset are available at: https://www.cvlibs.net/datasets/kitti/.

The DCP codes are available at GitHub:

- https://github.com/WangYueFt/dcp.

The IDAM codes are available at GitHub:

- https://github.com/jiahaoli95/idam.

The IDAM codes are available at GitHub:

- https://github.com/supersyq/RIENet.

## Supplemental Information

Supplemental information for this article can be found online at http://dx.doi.org/10.7717/peerj-cs.1724#supplemental-information.

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
