# Peer review of "HTMC: hierarchical tolerance mask correspondence for human body point cloud registration"

_PeerJ Computer Science, doi:10.7717/peerj-cs.1724_

## Round 0.1 · original submission · Major Revisions

The reviewers found merit in the manuscript and recommended revision. I would also recommend a revision to address all the comments and suggestions. The revised manuscript will be subjected to re-review. Also, address the following minor comments

1. Numbering of the section heading is not continuous. Start numbering from Introduction itself.

2. Proofread the entire manuscript to remove typos and English language improvement.

3. Make your source code available online on GitHub or other suggested platform, and put its link in the manuscript.

Good luck

**Language Note:** The Academic Editor has identified that the English language must be improved. PeerJ can provide language editing services - please contact us at copyediting@peerj.com for pricing (be sure to provide your manuscript number and title). Alternatively, you should make your own arrangements to improve the language quality and provide details in your response letter. – PeerJ Staff

Reviewer 1 ·

Basic reporting

This manuscript proposes a Hierarchical Tolerance Mask Correspondence (HTMC) method to
achieve better alignment by tolerating obfuscation.

This manuscript is well-written and easy to follow.

Literature is covered as partial overlapping and Correspondence-based registration. It needs more coverage and related literatures.

Experimental design

- Define RMSE and MAE.
- Include F-score in table - 1 to 5.
- Compared methods like ICPBesl and McKay (1992) should cover in Related work section.

Validity of the findings

- Authors must provide the code repository link (Like, GitHub).
- Conclusion needs more detail.
- Authors mentioned that they produce a human body dataset for training a human body registration network in sub-section 3.2. Authors must provide details/steps for producing human body dataset.

Additional comments

Minor revision

Reviewer 2 ·

Basic reporting

Authors main contribution in this paper is to design a hierarchical tolerance correspondence approach tailored to point clouds featuring numerous similar local regions. And to differentiable mask for use in correspondence-based registration methods, which intelligently adjusts correspondence relations based on feedback from the results. Moreover, they produce a human body point cloud dataset specifically designed for training a model for human body point cloud registration. Over all clear professional English is used throughout the paper, however, for a layman the topic should be understandable enough. Therefore, i suggest to add background section that clearly explain all technical terms and scenario why HTMC is necessary and a background related to Partial overlapping registration and Correspondence-based registration.

Experimental design

A more technical detail is required of why use Euclidean distance whereas other statistical analysis are there. Add a table that clearly illustrate previous studies experimental design flaws, results analysis and compared the results

Validity of the findings

ok

---

## Round 0.2 · accepted · Accept

The authors have addressed all the comments and suggestions provided by the reviewers. The manuscript may be accepted for publication in the present form.

Reviewer 1 ·

Basic reporting

The paper can be accepted in the present form.

Experimental design

Authors have addressed all comments

Validity of the findings

It is well-addressed now by the authors

Additional comments

Accepted